# Hospital Wastewater—Source of Specific Micropollutants, Antibiotic-Resistant Microorganisms, Viruses, and Their Elimination

**DOI:** 10.3390/antibiotics10091070

**Published:** 2021-09-04

**Authors:** Tomáš Mackuľak, Klára Cverenkárová, Andrea Vojs Staňová, Miroslav Fehér, Michal Tamáš, Andrea Bútor Škulcová, Miroslav Gál, Monika Naumowicz, Viera Špalková, Lucia Bírošová

**Affiliations:** 1Department of Environmental Engineering, Faculty of Chemical and Food Technology STU, Slovak University of Technology in Bratislava, Radlinského 9, 812 37 Bratislava, Slovakia; tomas.mackulak@stuba.sk (T.M.); miroslav.feher@stuba.sk (M.F.); michal.tamas@stuba.sk (M.T.); andrea.skulcova@stuba.sk (A.B.Š.); 2Department of Nutrition and Food Quality Assessment, Faculty of Chemical and Food Technology STU, Slovak University of Technology in Bratislava, Radlinského 9, 812 37 Bratislava, Slovakia; klara.cverenkarova@stuba.sk; 3Department of Analytical Chemistry, Faculty of Natural Sciences, Comenius University in Bratislava, Ilkovičova 6, 842 15 Bratislava, Slovakia; andrea.stanova@uniba.sk; 4South Bohemian Research Center of Aquaculture and Biodiversity of Hydrocenoses, Faculty of Fisheries and Protection of Waters, University of South Bohemia in Ceske Budejovice, Zatisi 728/II, CZ-389 25 Vodnany, Czech Republic; 5Department of Inorganic Technology, Faculty of Chemical and Food Technology STU, Slovak University of Technology in Bratislava, Radlinského 9, 812 37 Bratislava, Slovakia; miroslav.gal@stuba.sk (M.G.); viera.spalkova@stuba.sk (V.Š.); 6Department of Physical Chemistry, Faculty of Chemistry, University of Bialystok, K. Ciolkowskiego 1K, 15-245 Bialystok, Poland; monikan@uwb.edu.pl; 7Department of Zoology and Fisheries, Faculty of Agrobiology, Food and Natural Resources, Czech University of Life Sciences Prague, Kamýcka 129, 165 00 Praha, Czech Republic

**Keywords:** hospital wastewater treatment, pharmaceuticals, antibiotic-resistant microorganisms, antimicrobial resistance genes, SARS-CoV-2

## Abstract

Municipal wastewaters can generally provide real-time information on drug consumption, the incidence of specific diseases, or establish exposure to certain agents and determine some lifestyle consequences. From this point of view, wastewater-based epidemiology represents a modern diagnostic tool for describing the health status of a certain part of the population in a specific region. Hospital wastewater is a complex mixture of pharmaceuticals, illegal drugs, and their metabolites as well as different susceptible and antibiotic-resistant microorganisms, including viruses. Many studies pointed out that wastewater from healthcare facilities (including hospital wastewater), significantly contributes to higher loads of micropollutants, including bacteria and viruses, in municipal wastewater. In addition, such a mixture can increase the selective pressure on bacteria, thus contributing to the development and dissemination of antimicrobial resistance. Because many pharmaceuticals, drugs, and microorganisms can pass through wastewater treatment plants without any significant change in their structure and toxicity and enter surface waters, treatment technologies need to be improved. This short review summarizes the recent knowledge from studies on micropollutants, pathogens, antibiotic-resistant bacteria, and viruses (including SARS-CoV-2) in wastewater from healthcare facilities. It also proposes several possibilities for improving the wastewater treatment process in terms of efficiency as well as economy.

## 1. Introduction

As a result of human activity, water pollution has become a global challenge. This raises significant concerns, in particular as regards the presence and risk of pharmaceuticals, other chemical compounds, and pathogenic microorganisms in wastewater. Water pollution is a consequence of industrial production, laboratory research, high consumption of medicines, and, most importantly, the existence of healthcare facilities. Healthcare facilities are continuous point sources of contamination by various types of pharmaceuticals or diagnostic agents, such as contrast media [1,2]. Concentrations of such compounds in surface waters are relatively low, but information on their potential long-term effect on living organisms is still unknown.

It is unrealistic to generally expect a decline in the production or use of some dominant drug groups (especially antidepressants, antibiotics, or medication for cardiovascular diseases) in the near future. However, it is necessary to find ways for health care providers and the public to minimize the overuse of pharmaceuticals. At present, the use of CRP (C-reactive protein) tests by general practitioners is insufficient in several parts of Europe, despite the fact that their more frequent use could to some extent reduce the consumption and subsequent presence of antibiotics in the environment [3]. The use of antibiotics has a growing trend in the long run and its stagnation or decline is not expected [4]. Research is already responding to this, pointing to the possible use of new types of pharmaceuticals in combination with nanotechnologies to combat the emergence of resistance. New technologies use various nanomaterials based on carbon, titanium, silver, or gold with antibacterial activity [5]. The possibility of modifying classic materials to obtain good antibacterial properties and their subsequent possible use in healthcare facilities, for example in the treatment of wall surfaces in waiting rooms, is also being investigated [6]. However, all these substances would be additional and problematic wastewater pollutants.

Contrast agents, especially iodine-based substances, can also be a significant environmental problem in the future. A limited overview of the use of such types of compounds also means that the professional public does not have a comprehensive summary of their contribution to surface water contamination and their impact on various components of the environment in Slovakia, the Central European region, or elsewhere in the world. Compared to pharmaceutical products, these are extremely biologically inert and have only been studied to a limited extent. Another problem is the fact that even some innovative technologies, such as ozonation, cannot completely remove them from wastewater [7].

The presence of pharmaceuticals in surface waters points to the need for applications of new types of technologies (they can also be various combinations such as membrane technologies and oxidation processes or sorption materials), which could remove resistant types of microorganisms from wastewater in addition to micropollutants.

Generally, wastewater treatment at municipal WWTPs is often economically demanding, due to the significant flow of wastewater during the day. Therefore, it would be necessary to focus more on the treatment of dominant point sources of this type of pollution, such as healthcare facilities in this case, where the wastewater flow can be in volumes of up to 5000 m^3^ per day. These waters are significantly more contaminated with contrast agents, pharmaceuticals, and their metabolites compared to municipal wastewaters [8].

It should be emphasized that the monitoring of contrast agents, pharmaceuticals, and their metabolites in surface or drinking waters is often limited in time and money and therefore unsystematic. It regularly takes place only in some regions of Europe, which is why the ability of scientists to compare individual regions in terms of the presence of these micropollutants is limited [9].

## 2. Wastewater from Healthcare Facilities

### 2.1. Presence of Specific Micropollutants

Pharmaceuticals, hormones, contrast media, legal and illegal drugs, and their metabolites can be found in wastewater from various healthcare facilities at significant concentration levels [10]. With the exception of Denmark, where discharge limits are specified, most countries do not have special regulations for the disposal of wastewater from healthcare facilities [1]. The fate of these compounds in the environment is still largely unknown and published studies often deal only with their monitoring in wastewater and surface water [11]. On the other hand, values of other parameters for wastewater from healthcare facilities such as COD, NH_4_, etc. are similar to municipal wastewater (Table 1) [9].

According to published data, healthcare facilities are also perceived as a possible source of analgesics, antidepressants, antibiotics, or antiepileptics, but also antibiotic-resistant bacteria [10]. Heberer and Feldmann published a study aimed at discovering the occurrence of selected pharmaceuticals in wastewater from healthcare facilities in Berlin [12]. They found that pharmaceuticals were removed only to a limited extent by the sewage system and the treatment plant, with most of them entering surface waters. The authors point out that the main sources of carbamazepine and diclofenac in Berlin’s wastewater are predominantly households and hospitals with more than 12,000 beds. Saussereau et al. analyzed the hospital effluent in Rouen, France, and the results were then compared with the influent and effluent from a treatment plant [13]. Twenty frequently used pharmaceuticals were monitored in wastewater from healthcare facilities. The compounds with the highest measured concentrations were tramadol, venlafaxine, citalopram, caffeine, and oxazepam. Concentrations of these drugs in wastewater ranged from 0.1–2.4 μg·L^−1^.

Wastewater analyses in selected healthcare facilities in Slovakia, specifically in Bratislava, pointed to high amounts (showed elevated concentrations compared to municipal wastewaters) of pharmaceuticals such as tramadol (opioid analgesic) and midazolam (sedative) (Table 2). We observed an increased incidence of methamphetamine in wastewater from the Ružinov polyclinic, which may be related to the treatment of drug-addicted patients abusing this addictive substance. On the other hand, levels of cocaine, LSD, and MDMA occurred only to a limited extent, in concentrations below LOQ in this type of wastewater. Among legal drugs, caffeine predominates (bellow LOQ) in wastewater from healthcare facilities. Increased nicotine use results in increased concentrations of its metabolite cotinine (bellow 6700 ng/L).

The authors Yuan et al. (Table 3) monitored 22 psychoactive drugs in the effluent from two psychiatric hospitals in Beijing [14]. The pharmaceuticals with the highest concentrations in wastewater were representatives of neuroleptics: clozapine, quetiapine (antipsychotic medication used for the treatment of schizophrenia); sulpiride (a benzamide neuroleptic used in the treatment of schizophrenia and other psychotic disorders); antidepressants: fluvoxamine, citalopram (selective serotonin reuptake inhibitors indicated for depression and symptoms of depressive disorders); but also the popular carboxamide derivatives in our country: carbamazepine (antiepileptic drug) and benzodiazepine: oxazepam (anxiolytic).

### 2.2. Presence of Antibiotic-Resistant Microorganisms

Antibiotic and biocide-resistant bacteria have become a global challenge, which is slowly shifting our society into the so-called post-antibiotic era. European Commissioner for Health Stella Kyriakides said that in current pandemic times, COVID-19 and antimicrobial resistance call for a united approach across policies, countries, and all levels of society [15,16]. Many countries and global organizations are working to address this through various monitoring programs and measures (the One Health Action Plan in the EU, for example). The development and spread of antibiotic resistance is a complex process that involves many variables [17]. This phenomenon is largely influenced by the selection pressure induced by the presence of antimicrobials at subinhibitory concentrations. Pharmaceutically active compounds induce the formation of mutations leading to antibiotic resistance in bacteria. Some pharmaceuticals have been shown to increase the rate and likelihood of resistance genes transmission from resistant bacterial strains to susceptible ones, thereby contributing to the spread of resistance [18,19].

Wastewater from healthcare facilities, the source of various sensitive or antibiotic-resistant bacteria, is often discharged into the sewage system without prior treatment, thus contributing to an increase in the concentration and spectrum of pharmaceuticals as well as the number of antibiotic-resistant bacteria and antibiotic-resistance genes in municipal wastewater [20,21]. According to the data from the European Center for Disease Control (ECDC), at least one in three hospitalized patients and one in two patients undergoing surgery in the EU receives antibiotic treatment on any given day. Some of these uses may be unnecessary and may contribute to the spread of antimicrobial resistance [22]. Many hospitalized patients are infected with resistant bacteria or can be asymptomatic carriers of such bacteria. For example, a Europe-wide sequencing survey of 2000 samples of *Klebsiella pneumoniae* from patients in 244 hospitals in 32 countries showed that the hospital environment provides suitable space for the transfer of genes encoding carbapenemases (enzymes that cleave the latest generation of cephalosporins) [23]. In addition, the microbiota and microbiome of healthy individuals contain antibiotic-resistant bacteria and resistance genes [24]. The degree of bacterial resistance in healthcare facility effluents can be significantly different compared to other aquatic environments due to the use of specific antimicrobial agents in healthcare facility conditions. These are, for example, cefotiam, piperacillin, and vancomycin, which are used exclusively in the environment of healthcare facilities. Wastewater from healthcare facilities is therefore an important source not only of pharmaceuticals but also of antibiotic-resistant bacteria and resistance genes. Thus, it represents a very suitable environment for the development and spread of the antimicrobial resistance phenomenon [21,25,26,27].

Effluents from healthcare facilities in Slovakia and the Czech Republic contain relatively high levels of coliform bacteria (including *E. coli*) and gram-positive enterococci with the majority showing a multidrug resistance phenotype [25,28,29]. Such strains present in untreated wastewater from healthcare facilities can enter municipal WWTPs and, despite the high degree of dilution in the sewerage system, pass through the WWTP to the recipient and the environment [30]. Although antibiotic-resistant bacteria do not always pass through WWTPs, many antibiotics do and reach surface waters thus contributing to the development of antibiotic resistance. The numbers of antibiotic-resistant coliform bacteria in the wastewater of Slovak and Czech healthcare facilities are about one logarithmic order higher compared to the numbers at the WWTP inflow. The numbers of antibiotic-resistant *E. coli* and enterococci are at a similar level or slightly reduced compared to inflow waters [28,31]. However, hospital wastewater contains a high number of multidrug-resistant coliform bacteria and *E. coli*, we must consider the risk of the occurrence of strains producing broad-spectrum beta-lactamases (ESBLs). These strains can transfer multiple resistance genes through the conjugative plasmid and spread them to susceptible bacterial species. A massive occurrence of ESBL-producing bacteria has been recorded in wastewater from healthcare facilities. In addition, isolates from healthcare facility outlets are often characterized by carrying several different resistance genes on plasmids or chromosomes at the same time, making them able to resist a wide range of different antimicrobials [25,32,33].

In addition to ESBL-producing coliform bacteria, outlets from healthcare facilities are also considered to be the primary reservoir of vancomycin-resistant enterococci, which also belong to clinically relevant groups of bacteria. However, the monitoring of wastewater in Slovakia showed that these resistant’s occurred in municipal wastewater rather than in the effluents from healthcare facilities [28,31]. Another important issue is that low concentrations of non-antimicrobial pharmaceuticals such carbamazepine, atenolol, valsartan but also cotinine in wastewater can also contribute to the development of antibiotic resistance. However, hospital effluents contain a mixture of different pharmaceuticals at low concentrations, they can also significantly contribute to the development and spread of antibiotic and biocide-resistant bacteria and resistance genes [26].

### 2.3. Presence of Viruses in Wastewater from Healthcare Facilities

Nowadays, the most discussed topic is the presence of viruses in the wastewater due to the SARS-CoV-2 pandemic and their proven presence in municipal wastewater. However, the investigation of viruses in medical facilities wastewater is a topic of broader importance. Collected sewage from health facilities is abundant in large amounts of toxic substances and pathogenic organisms such as bacteria and viruses. Therefore, insufficient degradation of the medical wastewater might lead to pathogen circulation in the environment and recurring infections [34]. The most common viruses that can be found in the treated wastewater are enteric viruses such as Hepatitis A, but also noroviruses, rotaviruses, adenoviruses, and astroviruses. These cause a variety of diseases such as gastroenteritis, meningitis, and hepatitis. Most of them replicate massively in the intestinal tract and are shed into the feces in high concentrations [35].

SARS-CoV and SARS-CoV-2 were likewise detected in the hospital wastewater [36,37]. It was proven that SARS-CoV-2 binds to angiotensin-converting enzyme 2 (ACE2) located on the surface of enteric cells, replicates, and is shed into the feces similarly to enteric viruses. The fact was subsequently employed by many researchers and public health authorities to detect SARS-CoV-2 by RT-qPCR in the wastewater and used as an epidemiological tool for controlling the local SARS-CoV-2 epidemic [38]. Although reinfection by SARS-CoV-2 from wastewater has not been observed yet, further research is necessary as we know that many other viruses are found infectious in wastewaters including SARS-COV-1, and the presence of the SARS-CoV-2 was already detected in surface waters [39,40,41,42]. Cycle thresholds (CT) vary and reflect the pandemic situation in the population. Lately, the investigation involves genetic sequencing as a potential method for detecting mutations in the wastewater. The method will be even more crucial to employ at healthcare facilities as vaccination and the convalescence plasma treatment will increase selective pressure on the virus to mutate [43]. Analysis of new mutations and variants will again help epidemiologists in setting the right public health measures to contain the pandemic. Similarly, new mutations lead to higher infectivity of the virus, with fewer virus particles required for infection, which can play an important role in wastewater reinfections [44].

After the precise detection of viruses in the hospital wastewater, effective elimination is needed. Chlorination, ultraviolet, and ozone treatment are all commonly used disinfection technologies in hospital wastewater treatment plants and are also proven to be effective in the case of virus elimination [45]. On the contrary, a study by Zhang et al., reported detection of RNA SARS-CoV-2 in septic tanks after disinfection with sodium hypochlorite according to WHO guidelines [46]. Therefore, a combination of chlorination and other disinfection technologies such as plasma treatment, advanced oxidation reaction (e.g., Fenton reaction), ultrafiltration, or adsorption on nanoparticles is vital to avoid reinfections [47].

## 3. Innovative Processes Efficient in the Treatment of Wastewater from Healthcare Facilities

Current research is focused not only on monitoring the known pharmaceuticals and their metabolites in wastewater but also on studying the potential of new degradation processes. The study by Yuan et al. dealt with the ability of treatment plants to remove these types of micropollutants from the wastewater and to decrease the load on the environment [14]. The results showed only a limited ability to treat wastewater using biological processes (activated sludge) implemented in treatment plants. The complex structures of some compounds (especially pharmaceuticals with two aromatic rings) are highly resistant to biological purification, leading to a limited ability to degrade them. The lowest removal efficiency was obtained for oxazepam. In the case of sulpiride and carbamazepine, we even observed an increase in their concentration in the treatment plant from psychiatric hospital B. The collection time, the retention time, the sorption/desorption in the treatment plant, and the physicochemical properties of the studied compounds influence the concentration of the compounds [14]. An experimental technique based on the sub- and super-critical water oxidation of wastewater was used for amoxicillin and ciprofloxacin elimination. The feasibility of the method was tested in the temperature range from 473 K to 773 K and at flow rates of 3 and 5 mL/min. The highest COD and TOC reductions were achieved at the highest temperature of 773 K, where they were reduced by 76% and 63%, respectively [48]. However, conventional treatment (based mainly on mechanical and biological processes) is not efficient enough [49,50,51]. New treatment processes (see examples in Table 4) include various membrane bioreactors, nanomaterials, ferrates, Fenton reaction, ozonation, heterogeneous catalysis, ultrasound, aquatic plants, and adsorption e.g., on biochar or activated carbon [51,52,53,54].

Hospital wastewaters often contain significant amounts of fecal coliforms, which exhibit resistant or multi-resistant properties to various types of antibiotics [52,53,58,59,60]. Therefore, the degradation processes are investigated in terms of removal efficiency not only for a wide range of micropollutants but also for pathogenic microorganisms, so the disinfection ability is evaluated [61,62].

Currently, many different methods for the treatment of wastewater from healthcare facilities are being researched, which often combine chemical and biological degradation procedures in different ways [63,64,65]. A combination of ozonation with active sorbent (e.g., activated carbon), UV-C and H_2_O_2_, MBR (membrane bioreactor), and AOPs (advanced oxidation processes) dominates [52,66,67,68,69]. The historically used methods for disinfection of these types of waters are chlorination and UV-C radiation [69,70]. However, the main drawback of chlorination is the risk of the formation of various chlorine-rich organic by-products, which can negatively affect water organisms [52]. As in municipal wastewater, wastewater treatment from healthcare facilities can be performed by conventional chemical (e.g., coagulation) [71], and biological processes—where nitrification [52] predominates. In addition, research can be observed in various innovative processes such as the applicability of the enzymes themselves [72] or their combination with a root treatment plant [63], the combination of activated sludge using vermifiltration [65], the applicability of wood-destroying fungi [20], or various modifications of nanomaterials, sorbents and related e.g., photocatalysis (Table 5) [61,73].

Advanced oxidation processes (AOPs) and their various combinations with biological processes [67,74] achieve the best efficiencies in the treatment of wastewater in terms of chemical and biological pollution. An essential step in the effectiveness of AOPs is the production of free radicals, where the hydroxyl radical and various forms of reactive oxygen species (ROS) predominate [52,53,59,71,75,76,77,78]. AOPs that are frequently investigated include electrochemical AOPs [75,78,79,80].

In general, the most effective AOPs that are currently being intensively investigated in terms of wastewater treatment include the Fenton reaction (FR), photo-Fenton reaction(pFR), ozonation, and their modifications [29,52,53,75,76,77,81,82]. The combination of FR with biological processes also appears to be interesting [58,59]. The combination of a Fenton or photo-Fenton reaction, followed by purification of the effluent from a healthcare facility using activated sludge has been described in a study by Kajitvichyanukul et al. [58]. In a study by Miralles-Cuevas et al., nanofiltration was placed before the photo-Fenton reaction [83]. In addition to already described procedures, and the abilities of ferrates-iron (VI)—use in water purification and disinfection [53] are becoming a topical issue. Ferrates are very strong oxidizing agents that are able to degrade a wide range of drugs present in wastewater [84,85,86]. Currently, there is limited information from published literature about their disinfection power and purification of concentrated point sources of micropollutants such as healthcare facilities.

Current advances in the development of new technologies and materials in the degradation of micropollutants (pharmaceuticals, drugs, pesticides) in wastewater offer the use of boron-doped diamond electrodes [53]. The advantage of these electrodes is the significant increase in wastewater disinfection efficiency, as their application generates radical forms of oxygen (singlet oxygen, hydroxyl radical).

## 4. Conclusions and Suggestions

Various analytical procedures are used to monitor the presence and concentration of micropollutants, which may ultimately be reflected in an inaccurate description of the current situation. As part of a systematic collection, it would be interesting to obtain results directly from the WWTP (inflow and outflow). However, placing analytical equipment (mostly LC-MS/MS) in treatment plants seems unrealistic for application for several reasons. The price of the instrument, its operation and the need of a specially trained analytical chemist dominate. Therefore, it would be necessary to have central laboratories in different regions of Europe to regularly collect and evaluate water samples to monitor micropollutants.

After precise monitoring, efficient degradation should take place. Currently, hospital wastewaters are treated with a combination of physical and chemical processes. Membrane technologies, sorbents, UV irradiation, and chlorination are the most commonly used but oftentimes insufficient in degrading complex micropollutants and organisms. These should be further combined with AOPs for the successful treatment of such wastewater for alleviating the burden on the environment.

Another possibility is the development and utilization of novel (bio)sensors that can continuously monitor the selected micropollutants in wastewater and transfer the recorded data wirelessly to the monitoring station. There, the situation will be assessed and, if necessary, steps can be taken to eliminate the current environmental burden.

It would also be necessary to improve the life cycle of the released pharmaceuticals. Targeted consumer motivation can be an example of how to solve this problem so that it does not end in improper disposal, for example in sewers (after the expiration date or the patient’s death).

The last, but not the least, possibility is the improvement of treatment technologies based on AOPs. In combination with automatic monitoring systems present directly in the sewer or at the wastewater treatment plant, the wastewater treatment processes with AOPs can be started automatically and instantly, and thus react immediately to the unfavorable environmental situation.

## Figures and Tables

**Table 1 antibiotics-10-01070-t001:** Basic parameters of wastewater from hospitals and healthcare facilities [9].

Parameters		Range of Values
Water quality	pH	6–9
	Redox potential (mV)	850–950
	Conductivity (μS·cm^−1^)	300–1000
	Chlorides (mg·L^−1^)	80–400
	Nitrogen (mg N_2_·L^−1^)	60–98
	NH_4_ (mg NH_4_·L^−1^)	10–68
	Nitrites (mg NO_2_·L^−1^)	0.1–0.58
	Nitrates (mg NO_3_·L^−1^)	1–2
	PO_4_ (mg P-PO_4_·L^−1^)	6–19
	Soluble compounds (mg·L^−1^)	120–400
	Oils (mg·L^−1^)	50–210
	COD (mg·L^−1^)	1350–2480
	TOC (mg·L^−1^)	31–180
	BOC_5_/CHSK	0.3–0.4
	AOX (mg·L^−1^)	0.55–100
Microorganisms	*E. coli*	10^3^–10^6^
	*Enterococci*	10^3^–10^6^
	Fecal coliforms	10^3^–10^4^
	Total coliforms	10^5^–10^7^
	EC_50_ (*Daphnia*), TU	9.8–117
Organics	Total disinfective substances (mg·L^−1^)	2–200
	Total antibiotics (mg·L^−1^)	0.03–0.2
	Cytostatics (mg·L^−1^)	0.005–0.05
	Lipides regulators (mg·L^−1^)	0.001–0.01
	*Beta*-blocators (mg·L^−1^)	0.0004–0.025

**Table 2 antibiotics-10-01070-t002:** Composition of wastewater in selected Bratislava healthcare facilities with emphasis on the presence of specific micropollutants (pharmaceuticals and drugs) [9].

Substance	DFNsP	UNB Petržalka	Polyclinic Ružinov
		(ng·L^−1^)	
Caffeine	<LOQ	<LOQ	<LOQ
Cotinine	1100	280	6700
Codeine	21	<LOQ	10
Amphetamine	<LOQ	<LOQ	190
Oxycodone	<LOQ	<LOQ	<LOQ
Methamphetamine	28	25	1100
MDMA	<LOQ	<LOQ	<LOQ
Norketamine	<LOQ	<LOQ	<LOQ
Mephedrone	<LOQ	<LOQ	<LOQ
Ketamine	18	29	<LOQ
Benzoylecgonine	<LOQ	<LOQ	<LOQ
Tramadol	260	510	2400
Cocaine	<LOQ	<LOQ	<LOQ
LSD	<LOQ	<LOQ	<LOQ
Venlafaxine	75	<LOQ	600
Oxazepam	38	<LOQ	52
Citalopram	173	47	250
Midazolam	680	18	<LOQ
Buprenorphine	<LOQ	<LOQ	<LOQ
EDDP	<LOQ	<LOQ	<LOQ
Methadone	<LOQ	<LOQ	<LOQ
THC-COOH	52	<LOQ	<LOQ
Terbutaline	15	240	20
Atenolol	<LOQ	160	<LOQ
Bisoprolol	42	320	5200
Ampicillin	<LOQ	<LOQ	<LOQ
Penicillin V	<LOQ	<LOQ	<LOQ
Clonazepam	<LOQ	<LOQ	<LOQ
Atorvastatin	12	40	294
Flumequine	<LOQ	<LOQ	<LOQ
Metoprolol	96	310	2600
Ranitidine	31	1400	32
Furosemide	450	340	560

**Table 3 antibiotics-10-01070-t003:** Average drug concentrations determined in the effluent from psychiatric hospitals (A, B) and at the effluent of the relevant WWTP [14].

Substance (ng·L^−1^)	Effluent Psych. Hospital A	Secondary Effluent from WWTP	Effluent Psych. Hospital B	Secondary Effluent from WWTP
Clozapine	5600	300	5000	1200
Oxazepam	940	750	290	190
Sulpiride	2800	430	9800	11,000
Quetiapine	2000	<LOQ	5000	1200
Citalopram	67	19	260	160
Carbamazepine	88	<LOQ	160	180

**Table 4 antibiotics-10-01070-t004:** Technological processes and their combinations in the treatment of wastewater from healthcare facilities [9,51,52,53,55,56,57].

Treatment Process	Aim
Ozonation	Disinfection/degradation
Chlorination	Disinfection
Photo-Fenton reaction	Disinfection/degradation
Fenton reaction and modifications	Disinfection/degradation
Coagulation—filtration—disinfection	Disinfection/degradation
Ozonation/UV radiation	Disinfection/degradation
Ozonation/UV radiation/H_2_O_2_	Disinfection/degradation
Ozonation/UV radiation/H_2_O_2_/biological degree	Disinfection/degradation
Septic/anaerobic filter	Degradation
Septic/Fenton reaction	Disinfection/degradation
Flocculation/activated sludge	Degradation
Anaerobic and aerobic reactor with stabilized biofilm	Degradation
Aerobic reactor with stabilized biofilm/ozonation	Disinfection/degradation
Activated sludge	Degradation
Activated sludge/chlorination	Disinfection/degradation
Bioreactor—filamentous fungi	Degradation
Membrane bioreactor (MBR)	Degradation
MBR in combination with sorbents, AOPs, chlorination, catalysis	Disinfection/degradation
BDD—boron-doped diamond electrode	Disinfection/degradation
Ferrates (Fe^6+^) Anodic Oxidation with solid polymer electrolyte Ultrasound irradiation	Disinfection/degradation Disinfection/Degradation Disinfection/Degradation

**Table 5 antibiotics-10-01070-t005:** Monitoring of the occurrence of specific micropollutants in effluents from healthcare facilities after the treatment process.

Compound	Effluent Concentration (µg·L^−1^)	Study
Caffeine	12.3–42	[87]
	15.6	[88]
	12.1–182	[89]
	<7.2	[53]
Carbamazepine	0.03–0.07	[90]
	<0.017–1.7	[87]
	LOD–0.24	[14]
	0.7–2.7	[91]
	0.64–1.2	[92]
	0.222	[93]
	0.018–6.08	[89]
	0.003–0.036	[94]
	0.163	[88]
Citalopram	0.019–0.322	[14]
	47–490	[53]
Cocaine	0.05	[95]
	<19	[53]
Benzoylecognine (*metabolite cocaine*)	0.029	[95]
	<7	[53]
Codeine	0.378	[95]
	0.01–5.7	[90]
	0.26–3.2	[92]
	<2.3–58	[53]
6-acetylcodeine	<0.002	[95]
Diazepam	<0.001–0.038	[92]
	0.069	[93]
Ketamine	0.206	[95]
	<4.2–29	[53]
Lorazepam	0.17–0.79	[92]
	LOD–0.353	[14]
Lidocaine	9.133	[93]
Methamphetamine	0.26	[95]
	<4.2–1100	[53]
Morphine	1.24	[95]
	3.679	[93]
6-acetylmorphine	<0.0005–0.039	[95]
Oxazepam	0.186–0.942	[14]
	1.123	[93]
	<24–52	[53]
Tramadol	0.958	[14]
	260–2400	[53]
Venlafaxine	0.811	[14]
	<24–600	[53]

## Data Availability

Not applicable.

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
