# Peer review of "Hospital Wastewater—Source of Specific Micropollutants, Antibiotic-Resistant Microorganisms, Viruses, and Their Elimination"

_antibiotics, 2021, doi:10.3390/antibiotics10091070_

Round 1

Reviewer 1 Report

 On account of the manuscript ANTIBIOTICS-1321558, entitled “Hospital Wastewater – A Potential Source of Specific Micropollutants, Antibiotic Resistant Microorganisms and SARS-Cov-2” by Tomáš Mackuľak et al., the authors reviewed the recent knowledge for micropollutants, pathogens, antibiotic resistant bacteria, and SARS-CoV-2 in wastewater from healthcare facilities, and then discussed the possibilities to improve wastewater treatment process. The topic is important to better understanding of the impacts and to conduct the environmental risk management of medical wastewater, and the authors got interesting results. After careful consideration, I made a decision that the manuscript is acceptable for publication in its present form.

Special remarks:
‧ The present short review summarized the recent knowledge from studies dealing with micropollutants, pathogens, antibiotic resistant bacteria, and SARS-CoV-2 in wastewater from healthcare facilities, and addressed the possibilities to improve wastewater treatment process in term of efficiency as well as economy.
‧ Occurrence of micropollutants, pathogens, antibiotic resistant bacteria, and SARS-CoV-2 in hospital wastewater was urgent issue for both environmental and human health risk management. On the other hand, the knowledge was still limited worldwide.
‧ The present review presented wide range knowledge of emerging concerns in hospital wastewater and provided useful future prospects to better understandings for the management of hospital wastewater.
‧ The interpretation of the evidence and arguments presented and conclusions are sufficient.
‧ The references cited relevant and up to date.
‧ The tables and/or figures are useful, necessary, and good quality. 

Author Response

We would like to thank Reviewer 1  for his comments.

Reviewer 2 Report

The general topic is of interest, but the review is not well focused or organized. There are lots of mentions of topics that are not directly related to the main point of the review. There are topics covered (e.g. wastewater treatment methods) which take up a fair portion of the article but aren't mentioned in the title. There seems to be a lot of "fluff", either sentences that don't say a lot or have a lot of "hype", making things seem more significant than they really are. Since people are not generally exposed to hospital wastwater, the real issues are a) is the presence and abundance of these pollutants impacting the ability of WWTPs to remove them, and b) are there widespread examples of pollutants escaping treatment and appearing in receiving waters, and are is there actual evidence of consequences?  This review could use extensive revision with sharper focus on areas of importance. There is very little information on SARS-CoV2 and it seems to appear in the text and title to make the article seem more topical.  

Specific comments:

Hospital Wastewater – A Potential Source of Specific Micropollutants, Antibiotic Resistant Microorganisms and SARS-Cov-2

  1. 1. Abstract: some wording vague. English awkward in places.
  2. CRP undefined line 54.
  3. lines 58-64. Nanomaterials and wall surfaces? This is not a review on infection control in hospitals. Antimicrobial nanoparticles would be an additional and problematic wastewater pollutants.
  4. Veterinary? Relevant to topic? Relevant to scale? lines 65-72
  5. line 117 increased as compared to what?
  6. Table 2 is not clear. Does "<5.7" mean that 5.7 is the limit of detection and the value measured was below the detection limit?
  7. Especially given the title, would the presented information be better divided into specific categories? In other words, discuss antibiotics separately from other drugs, then bacteria, then viruses?
  8. Table 4 is confusing. Is this a list of methods as well as all the possible combinations of methods? The legend says "possible"; is there evidence of efficacy for all? The headings are poorly chosen: Why are the methods "Degradation Procedures" when some of the outcomes are only disinfection, which doesn't have a very specific scientific meaning anyway. If "disinfection" means kills microbes and "degradation" means breaks down organic pollutants, this should be made more clear.
  9. lines 168-182. The point of this paragraph is not clear. The stated problem is that hospital wastewater pollutants are not being adequately removed/destroyed. If different methods are being discussed, it should be made clear how these will solve the problem that treatments of municipal water are failing at.
  10. lines 202-206 . A major part of this review appears to be potential methods for solving the problem. This is not reflected in the title.
  11. lines 221-223. References used to state that transfer of antibiotic resistance genes was promoted by presence of pharmaceutical pollutants. This would seem like a very important finding, however that conclusion was not stated in the abstracts of either of the two papers listed.
  12. lines 250-252. Other sources indicate that antibiotic resistant bacteria ARE successfully removed during wastewater treatment. While an increase in antibiotic resistant bacteria in WWTPs is not good, nor is the escape of antibiotics into receiving waters, there does not appear to be a major problem with the escape of resistant bacteria to the environment.
  13. Lines 303-308. Detection of SARS RNA is not always indicative of the presence of whole, infective virus, and this study should be taken with caveats. It would also be worth mentioning RNA concentrations, perhaps as cycle number from RT-PCR results. Since fecal-oral transmission hasn’t been described, nor has escape of SARS CoV2 into receiving waters to my knowledge, this does not seem like a known means of reinfection and is not yet a problem needing to be solved.

Author Response

First of all, we would like to thank Reviewer2 for his comments, which contributed to better quality of our manuscript. 

The general topic is of interest, but the review is not well focused or organized. There are lots of mentions of topics that are not directly related to the main point of the review. There are topics covered (e.g. wastewater treatment methods) which take up a fair portion of the article but aren't mentioned in the title.

Response: Title was changed to cover the issue of wastewater treatment methods.

Specific comments:

Abstract: some wording vague. English awkward in places.

Response: English was edited.

CRP undefined line 54.

Response: CRP was defined.

lines 58-64. Nanomaterials and wall surfaces? This is not a review on infection control in hospitals. Antimicrobial nanoparticles would be an additional and problematic wastewater pollutants.

Response: Explanation was added. All these substances would be an additional and problematic wastewater pollutants

Veterinary? Relevant to topic? Relevant to scale? lines 65-72

Response: Statement was removed

line 117 increased as compared to what?

Response: Statement was corrected.

Table 2 is not clear. Does "<5.7" mean that 5.7 is the limit of detection and the value measured was below the detection limit?

Response: Table 2 was corrected.

Especially given the title, would the presented information be better divided into specific categories? In other words, discuss antibiotics separately from other drugs, then bacteria, then viruses?

Response: Structure of the manuscript was corrected according to reviewer proposal.

Table 4 is confusing. Is this a list of methods as well as all the possible combinations of methods? The legend says "possible"; is there evidence of efficacy for all? The headings are poorly chosen: Why are the methods "Degradation Procedures" when some of the outcomes are only disinfection, which doesn't have a very specific scientific meaning anyway. If "disinfection" means kills microbes and "degradation" means breaks down organic pollutants, this should be made more clear.

Response: Legend of table was corrected.

lines 168-182. The point of this paragraph is not clear. The stated problem is that hospital wastewater pollutants are not being adequately removed/destroyed. If different methods are being discussed, it should be made clear how these will solve the problem that treatments of municipal water are failing at.

Response: In paragraph was added statement made it more clear.

lines 202-206 . A major part of this review appears to be potential methods for solving the problem. This is not reflected in the title.

Response: Title was changed to cover the issue of wastewater treatment methods.

lines 221-223. References used to state that transfer of antibiotic resistance genes was promoted by presence of pharmaceutical pollutants. This would seem like a very important finding, however that conclusion was not stated in the abstracts of either of the two papers listed.

Response: References were corrected.

lines 250-252. Other sources indicate that antibiotic resistant bacteria ARE successfully removed during wastewater treatment. While an increase in antibiotic resistant bacteria in WWTPs is not good, nor is the escape of antibiotics into receiving waters, there does not appear to be a major problem with the escape of resistant bacteria to the environment. Response: Statement was corrected

Lines 303-308. Detection of SARS RNA is not always indicative of the presence of whole, infective virus, and this study should be taken with caveats. It would also be worth mentioning RNA concentrations, perhaps as cycle number from RT-PCR results. Since fecal-oral transmission hasn’t been described, nor has escape of SARS CoV2 into receiving waters to my knowledge, this does not seem like a known means of reinfection and is not yet a problem needing to be solved.

Response: An explanation of importance to monitor SARS-Cov-2 in wastewater has been added.

Reviewer 3 Report

English needs improving. Introduction is caotic; the authors need to identify and discuss separately different pollutants/different technologies, avoiding repeating generic statements regarding the expected concern or the not expected decrease of the use. Furthermore, section 2 should be part of the Introduction. The discussion of each technology in the core section ‘Innovative processes efficient for treating wastewater from healthcare facilities’ needs to be heavily expanded. Furthermore, criticism needs to be added to the discussion once introduced. Otherwise, the manuscript could not be considered as a review. Below additional comments 23 It can provide 26 real-time information about drug consumption, about specific diseases incidence or can establish exposure to certain agents and determine some lifestyle consequences.’ which can be true referring to wastewater from domestic/civil water treatment plants, is out of topic in the abstract, as the focus of the manuscript are treatment techs of wastewater form healthcare facilities. 45 ‘It.. It = what? Please replace with the subject 54 Please define here CRP and GP. Same applies for all acronyms in the text. 65 ‘The limited overview of the use of veterinary drugs’ Please rephrase 77 ‘Wastewater treatment at WWTPs is often economically demanding, due to the significant flow of wastewater during the day.’ does not apply to the focus of the manuscript, which is hospital wastewaters. Table 4 and related discussion: no mention of anodic oxidation with SPE and ultrasound irradiation. See 10.1016/j.jwpe.2019.101074 and https://doi.org/10.1007/s40201-020-00555-z 224 ‘Wastewater from healthcare facilities is an extremely important source of such sub-224 stances, as well as various sensitive or antibiotic-resistant bacteria.’ Please avoid continuous repletion of similar generic sentences out of the Introduction section 310 ‘Analytical procedures used to monitor the presence and concentration of micropol-310 lutants can often vary, which may ultimately be reflected in an inaccurate description of 311 the current situation.’ Please rephrase

Author Response

We would like to thank to Reviewer 3 for his comments which increase quality of our manuscript. 

English needs improving.

Response: English was edited.

Introduction is chaotic; the authors need to identify and discuss separately different pollutants/different technologies, avoiding repeating generic statements regarding the expected concern or the not expected decrease of the use. Furthermore, section 2 should be part of the Introduction. The discussion of each technology in the core section ‘Innovative processes efficient for treating wastewater from healthcare facilities’ needs to be heavily expanded. Furthermore, criticism needs to be added to the discussion once introduced. Otherwise, the manuscript could not be considered as a review.

Response: Whole manuscript was edited according to proposed comments.

It can provide 26 real-time information about drug consumption, about specific diseases incidence or can establish exposure to certain agents and determine some lifestyle consequences.’ which can be true referring to wastewater from domestic/civil water treatment plants, is out of topic in the abstract, as the focus of the manuscript are treatment techs of wastewater form healthcare facilities.

Response: Information was added.

‘It.. It = what? Please replace with the subject

Response: Statement was corrected

Please define here CRP and GP. Same applies for all acronyms in the text.

Response: Acronyms were defined.

 ‘The limited overview of the use of veterinary drugs’ Please rephrase

Response: Statement was removed.

‘Wastewater treatment at WWTPs is often economically demanding, due to the significant flow of wastewater during the day.’ does not apply to the focus of the manuscript, which is hospital wastewaters.

Response: Statement was corrected

Table 4 and related discussion: no mention of anodic oxidation with SPE and ultrasound irradiation. See 10.1016/j.jwpe.2019.101074 and https://doi.org/10.1007/s40201-020-00555-z 224 ‘Wastewater from healthcare facilities is an extremely important source of such substances, as well as various sensitive or antibiotic-resistant bacteria.’ Please avoid continuous repletion of similar generic sentences out of the Introduction section

Response: References were added, text and statement were corrected.

‘Analytical procedures used to monitor the presence and concentration of micropollutants can often vary, which may ultimately be reflected in an inaccurate description of  the current situation.’ Response: Statement was corrected.

Reviewer 4 Report

Manuscript - Hospital Wastewater - A Potential Source of specific Micropollutants, Antibiotic Resistant Microorganisms and SARS-Cov-2 has been peer-reviewed for publication in Antibiotics, and publication of the article is proposed after extensive – major revision.

In general, this review discusses the current issues in hospital wastewater treatment, and the article is generally well written and appropriately structured.

Comments: It is suggested to the authors to improve/correct the paper:

The abstract is too general.

A section on analytical methods is missing (only one sentence in the conclusions), this could be included in the introduction or in Chapter 2 - Table 2 or as a separate chapter.

Some discussion of modern processes such as sub- and supercritical water oxidation should also be included in Chapter 3 where innovative processes are discussed. Include the reference of Stavbar S. et al. Sub- and super-critical water oxidation of wastewater containing amoxicillin and ciprofloxacin. The Journal of supercritical fluids. [Print ed.] 2017, vol. 128, p. 73-78 and other similar references.

Author Response

We would like to thank Reviewer 4 for his comments for improvement of our manuscript.

The abstract is too general.

Response: Abstract was revised.

A section on analytical methods is missing (only one sentence in the conclusions), this could be included in the introduction or in Chapter 2 - Table 2 or as a separate chapter.

Response: Separate chapter about analytical methods needs more attention and should be separate review, however separate chapter would be to long.

Some discussion of modern processes such as sub- and supercritical water oxidation should also be included in Chapter 3 where innovative processes are discussed. Include the reference of Stavbar S. et al. Sub- and super-critical water oxidation of wastewater containing amoxicillin and ciprofloxacin. The Journal of supercritical fluids. [Print ed.] 2017, vol. 128, p. 73-78 and other similar references.

Response: information and discussion about sub- and supercritical water oxidation as well as citation was added in chapter 3.

Round 2

Reviewer 2 Report

This review is significantly improved and the authors are to be commended for their revision. I have only two remaining reservations, both relating to tables. The structure/labeling of Table 1 is still not satisfactory. The term "Biologicals" is unclear, with drugs being much different from microorganisms yest lumped together in the table. What I see in the table is water quality parameters, various organics such as antibiotics, and specific microbes. Grouping information under these three categories would make more sense.

Table 4 has a similar problem. The word "degradation" is being used much more broadly than is conventional in English. I strongly suggest that the column headed "Degradation" be replaced with "Treatment Process", as this can describe destruction, trapping, and removal methods, whereas "degradation" really refers only to destruction. Treatment Process also applies to disinfection methods as well.

Author Response

Labeling of Table 1 was changed.

legend of Table 1 and Table 4 was modified.

Reviewer 3 Report

Mnuscript has been sufficiently improved.

Author Response

We would like to thank reviewer 3 for the correction of our manuscript.